# FLASHLIGHT: ENABLING INNOVATION IN TOOLS FOR MACHINE LEARNING

## ABSTRACT

As the computational requirements for machine learning systems and the size and complexity of machine learning frameworks increases, essential framework innovation has become challenging. While computational needs have driven recent compiler, networking, and hardware advancements, utilization of those advancements by machine learning tools is occurring at a slower pace. This is in part due to the difficulties involved in prototyping new computational paradigms with existing frameworks. Large frameworks prioritize machine learning researchers and practitioners as end users and pay comparatively little attention to systems researchers who can push frameworks forward — we argue that both are equally-important stakeholders. We introduce Flashlight, an open source library built to spur innovation in machine learning tools and systems by prioritizing open, modular, customizable internals and state-of-the-art, research-ready models and training setups across a variety of domains. Flashlight enables systems researchers to rapidly prototype and experiment with novel ideas in machine learning computation and has low overhead, competing with and often outperforming other popular machine learning frameworks. We see Flashlight as a tool enabling research that can benefit widely-used libraries downstream and bring machine learning and systems researchers closer together.

## 1 INTRODUCTION

The recent rise of deep learning-based techniques has been accompanied and sustained by the wide availability of dedicated frameworks such as TensorFlow (Abadi et al., 2016) and PyTorch (Paszke et al., 2019). These frameworks have enabled the democratization of machine learning research by providing extensive collections of high level primitives to support common use cases. Lowering the barrier to entry for end users has boosted the popularity of both neural networks and the frameworks in which they are implemented. However, in order to support what are now vast ecosystems and a diverse user base, framework size and complexity have increased dramatically over time. As a result, deep, groundbreaking framework research has become extremely onerous and time consuming, precluding rapid innovation. Given these barriers, major deep learning frameworks have become stuck in their existing operating modes.

Innovation in this area remains as important as ever. Indeed, framework innovation accelerates machine learning (ML) and artificial intelligence (AI) research. Frameworks that are easier to use reduce the engineering burden on researchers, and frameworks that are higher-performance decrease the time required to iterate on experimental work and validate hypotheses. Even more critically, tooling plays a fundamental role in deciding which ideas succeed or fail. For example, LeCun et al. (1989) pioneered the use of convolutional neural networks (CNNs) (Fukushima & Miyake, 1982) trained using backpropagation for computer vision tasks in the late 1980s, which was subsequently applied to handwriting recognition. However, widespread success for CNNs was achieved two decades later when Krizhevsky et al. (2012a) leveraged the CUDA programming model to take advantage of graphics processing units (GPUs) to train a much deeper model (AlexNet).

While deep learning frameworks have been optimized to leverage existing hardware paradigms for common neural network architectures, they often fail to deliver similar efficiencies on designs that diverge from the mainstream. For example, Barham & Isard (2019) explain how the design of these frameworks results in poor hardware utilization for a novel type of neural network, known as a

capsule network (Hinton et al., 2018), that leverages new components such as squashing operations and routing by agreement. More generally, what are now unconventional approaches to modern problems in machine learning require highly-specialized additions to popular frameworks. As a result of narrowly-optimized systems, research beyond deep learning may be discounted due to purported computational infeasibility given modern frameworks' capabilities.

Furthermore, the waning of Moore's law (Theis & Wong, 2017) coupled with the ever-growing computational demands of deep learning are prompting several shifts in hardware. Massive-scale distributed computing is now required to train leading models — a process that established frameworks remain unable to handle truly automatically. In parallel, multiple specialized hardware products are now available to better support deep learning applications: Nvidia's TensorCores (Markidis et al., 2018), Google's TPUs (Jouppi et al., 2017), Graphcore's IPUs (Jia et al., 2019), Apple's Neural Engine[1], and others have been developed to improve total float-pointing operations (FLOPs), cost per-FLOP, or energy consumption. Additionally, numerous efforts are underway to move away from conventional von Neumann computing architectures in which memory and processing units are physically separated, either by storing data closer to compute units or by switching to in-memory computing altogether.

While tooling innovation is alive and well given these incentives for progress, working within large, well-established frameworks has become more and more challenging as framework size and scope grows. As a result, many recent innovations have required the development of ad-hoc tools. For example, efforts in machine learning-driven compilation of neural networks are largely built on top of Halide (Adams et al., 2019; Steiner et al., 2021) and TVM (Chen et al., 2018; Zheng et al., 2020); FlexFlow (Jia et al., 2018; 2020) underpins recent work aimed at improving the use of distributed computing to accelerate the training of large neural networks; and PET (Wang et al., 2021) provides a framework that enables graph-level neural network optimizations. With ad-hoc approaches, researchers are required to start from scratch for new directions or adapt their ideas to fit into the scaffolding these frameworks provide — resulting in significant technical burdens.

To sustain framework innovation, we introduce Flashlight, an open source minimalist ML library designed to support research in machine learning frameworks, facilitate rapid iteration on ideas, reduce the engineering burden on researchers, and remove the need for new tools. Flashlight includes:

- A modular, component-based architecture that makes every aspect of the implementation fully **customizable** with simple internal APIs.
- A compact yet highly-performant **reference implementation** of each component.
- A comprehensive set of **benchmarks** representative of the state-of-the-art in machine learning on which to evaluate alternative implementations.

## 2 RELATED WORK

Numerous frameworks have been implemented in recent years to support machine learning, including Lush (Bottou & LeCun, 2002), Theano (Bergstra et al., 2010), Torch (Collobert et al., 2011), Caffe (Jia et al., 2014), MXNet (Chen et al., 2015), deeplearning4j (Team, 2016), TensorFlow (Abadi et al., 2016), Flux (Innes, 2018), Jax (Bradbury et al., 2018), PyTorch (Paszke et al., 2019), Chainer (Tokui et al., 2019), and PaddlePaddle (Ma et al., 2019). These frameworks offer programming models designed around multidimensional arrays (TENSORS), modeled as first-class objects and supported by a comprehensive set of mathematical primitives (or operators) to manipulate them. To provide the computing power required by deep learning-based methods, most natively support hardware accelerators such as general-purpose GPUs or custom-designed ASICs such as TPUs.

Generally, framework implementations follow one of two computational models:

- In the *deferred execution* model, the neural network to be trained is first encoded as a dataflow graph which can be optimized for a specific set of target hardware devices. The neural network is then executed in a distinct second phase. Since the dataflow graph represents the entire computation, both local and global optimizations can be applied, making the subsequent execution very efficient. However, only programs that can be represented as

---

[1]https://nr.apple.com/dE9q1p9M7t

dataflow graphs can be processed with this approach, thus limiting flexibility. Frameworks such as Theano, TensorFlow[2], Caffe, or MXNet fall into this category.

- In the *eager* model, an interpreter (such as Python) is extended with the high level kernel-based operations needed to train a neural network. These operations are executed immediately when called, though this precludes many optimizations. By weaving neural network-related operations into a Turing complete programming language, this approach is extremely flexible. Furthermore, the imperative nature of the underlying programming language allows for fine-grained control over the execution order and memory utilization, which enables more specific user-driven optimization. Frameworks such as Torch, PyTorch, or Chainer exemplify this approach.

## 3 PRINCIPLES

The aforementioned frameworks are designed and implemented to best-serve their user bases — namely, machine learning researchers and practitioners. They rely on large, internally complex codebases to provide comprehensive solutions, as is further discussed in Section 5.

In contrast, Flashlight targets an audience of researchers interested in experimenting with new designs and implementations of machine learning tools or broader computational or modeling paradigms. To foster this type of innovation, Flashlight balances simplicity and nimbleness with the need to provide enough functionality to support real use cases. Internal and external simplicity is the key design principle of Flashlight; the ability to dramatically modify software and drive it in new directions is inversely correlated with codebase size and complexity (Gill & Kemerer, 1990). More specifically:

- Flashlight is built on a shallow stack of **idiomatic, modular, and customizable** abstractions. Framework components interact through small, well-defined, stable APIs, which expose most internal aspects of its implementation. This ensures that every component of Flashlight can be modified or replaced with new custom implementations, even e.g. its memory manager and tensor implementation. To support the exploration of a wide array of alternative approaches, Flashlight interfaces are **flexible and unopiniated** by design. This is in contrast to other frameworks, which impose stricter implementation requirements based on tight design constraints for their computation models and support requirements across hardware, downstream frameworks, and other ecosystem members.

- Flashlight provides deliberately-**compact default implementations** of its APIs. This reduces out-of-the-gate engineering burden and the need for modifications, and enables fast compilation and rapid iteration when experimenting. Furthermore, to mitigate premature optimization, Flashlight deliberately **abstains from adding small efficiency improvements** if they conflict with the goals of keeping the codebase simple and APIs clean.

- Flashlight is a **research-first** framework, and is not intended for out of the box production use. To keep codebase size small, it forgoes features such as model servers for deployment and integration with cluster management tools.

Flashlight is a viable solution for **machine learning research**, shipping with a comprehensive set of benchmarks and research setups for state-of-the-art neural network architectures such as convolutional neural networks (CNNs) (Krizhevsky et al., 2012b) and Transformers (Vaswani et al., 2017), as well as task-specific models such as ViT (Dosovitskiy et al., 2020), DETR (Carion et al., 2020), or BERT (Devlin et al., 2018). The speech recognition system wav2letter (Pratap et al., 2019), is also built entirely on Flashlight.

Benchmarks built on these state-of-the-art models make Flashlight a **turn key solution for system researchers** who want to quickly evaluate their design and implementation choices without needing to build test benches from the ground-up. More importantly, Flashlight makes possible end-to-end benchmarking on real models rather than microbenchmarks or small-scale tests.

---

[2]TensorFlow 2.0 adds support for eager execution semantics as well.

## 4 DESIGN

Flashlight's design is centered around *internal* APIs for framework components which form the building blocks for domain-specific ML *packages* and *applications* — this structure is outlined in Figure 1. Flashlight is implemented as a C++ library and follows a Tensor-based programming methodology, with neural network building blocks that derive from a MODULE interface, communicate by exchanging Tensor data, and are composed functionally or imperatively to form complete neural network architectures. Tensor programming in Flashlight is fundamentally dynamic, but given that C++ is a compiled language, code describing models in Flashlight is compiled. This approach promotes type safety, foregoes the runtime overheads associated with interpreters, and, unlike eager-based approaches, enables global optimizations where possible.

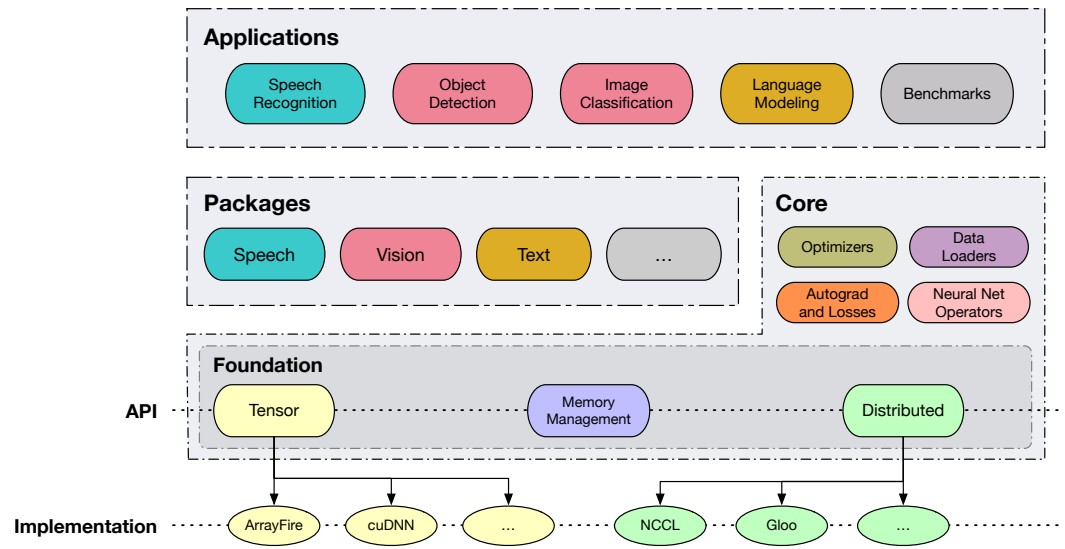

Figure 1: Components of the Flashlight library.

### 4.1 OPEN FOUNDATIONAL INTERFACES

Flashlight is built on top of three open *foundational* APIs, each addressing design and implementation challenges faced by machine and deep learning tools: a *Tensor* interface, a *memory management* subsystem, and a *distributed* computing interface. These APIs are backed by reference implementations that enable Flashlight to efficiently target CPUs, GPUs, and other accelerators. These include code generation and dedicated kernels for Intel, AMD, OpenCL, and CUDA devices, and leverage libraries such as cuDNN (Chetlur et al., 2014), MKL (Intel, 2020a), oneDNN (Intel, 2020b), ArrayFire (Malcolm et al., 2012), and MiOpen (Khan et al., 2019).

#### 4.1.1 TENSOR INTERFACE

Modern deep learning frameworks feature tensor library internals which sit under deep layers of abstractions, requiring numerous framework modifications in order to iterate on tensor stack design. Flashlight's TENSOR abstraction is defined in terms of existing tensor libraries via a simple, extensible interface and a high-level API that mirrors *numpy* (Harris et al., 2020) rather than using specific, opinionated intermediate representations (IRs) or large operator sets.

Flashlight TENSOR backend implementations need not follow any particular computation mode as outlined in Section 2 and shown in Figure 2. Tensor values need only be materialized upon user request — typically when extracting the output values of a model or inspecting intermediary state. This provides a flexibility unique amongst deep learning frameworks to either defer or eagerly-compute intermediate values — or to experiment with new computation paradigms altogether.

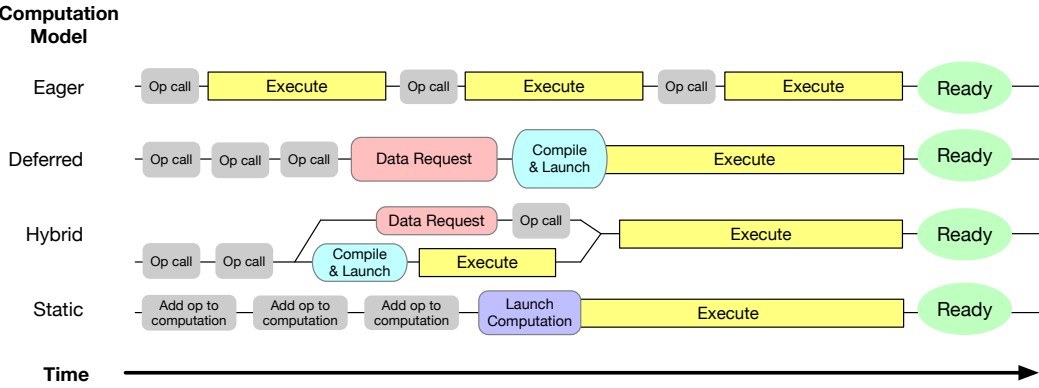

Figure 2: Flashlight's Tensor API supports backend implementations with any of the above computation modes.

Implementing a TENSOR backend in Flashlight involves fulfilling a small set of implementation requirements. Users have full control of their implementations after subclassing two interfaces:

- the TENSORADAPTER interface (Listing 1) which allows a backend to attach custom stateful information and metadata to each tensor. This includes shape, type, and memory information — which may be implementation-dependent.
- the TENSORBACKEND interface (Listing 2) which allows backends to store global state as needed (e.g device compute streams, dataflow graphs) and implement a small set of primitive tensor operations including unary and binary operations (e.g. arithmetic ops), reductions, matrix multiplication, and convolution.

Rather than require implementations of large, highly-specialized operator sets or interoperability with complex dispatch mechanisms or intermediate representations (IRs) as do other frameworks, Flashlight operators outside of the small TENSORBACKEND API are derived by composition. For example: the ReLU activation is implemented by leveraging the MAX operator. Flashlight's reference TENSOR implementation uses a *hybrid* approach, offloading computation to highly-optimized vendor libraries when advantageous and relying on deferred, on-the-fly code generation via ArrayFire for all other operations so as to increase kernel arithmetic intensity.

```
class MyTensorImpl : public TensorAdapter {
  // State information goes here (e.g. buffers, shape)
 public:
  // Metadata
  const Shape& shape() override;
  dtype type() override;
  // ...
};
```
Listing 1: The *TensorAdapter* interface for implementing operations on tensor metadata and storing tensor state for individual tensor instances.

```
class MyTensorBackend : public TensorBackend {
  // State information goes here (e.g. compute streams, compiler state)
 public:
  // Tensor operation primitives
  Tensor add(const Tensor& lhs, const Tensor& rhs) override;
  Tensor minimum(const Tensor& lhs, const Tensor& rhs) override;
  // ...
};
```
Listing 2: The *TensorBackend* interface for implementing operations on tensors and storing global backend state.

### 4.1.2 MEMORY MANAGEMENT

Robust memory management is an important research area as model size increases. While individual TENSOR backends in Flashlight can perform their own memory management as defined by implementers, Flashlight's default TENSOR backend also provides a generic API for defining custom memory management schemes. By default, memory is only allocated when needed for just-in-time compilation. A sample of this API is shown in Listing 3. To support the lazy computation model as well as just in time code generation, memory allocations only occur when tensors need to be materialized per the compute graph. Buffers are used asynchronously after they are requested depending on the timing of kernel launches but are not freed until computation is complete.

```
1 class CachingMemoryManager : public MemoryManagerAdapter {
2   // Store state as needed
3 public:
4   void* alloc(bool userLock, const unsigned ndims,
5               dim_t* dims, const unsigned elSize) override;
6   void unlock(void* ptr, bool userLock) override; // free memory
7   // ...
8 };
```
Listing 3: An implementation of a memory manager using the memory management API.

### 4.1.3 DISTRIBUTED TRAINING

Flashlight provides a low-level API for distributed training primitives with performant default implementations in GPU and CPU settings using NCCL (2019) and Gloo (2019), respectively. Users can add new backends or custom methods of performing distributed computation, and can use primitives in other framework components as needed. The API is unopinionated and supports both synchronous and asynchronous communication and arbitrary distributed computation schemes.

### 4.2 THE FLASHLIGHT CORE

The TENSOR API, together with memory management and distributed computation abstractions provide a foundation on which to build other core machine learning algorithms and applications. These other core components are outlined below. Section A.4 provides code samples and linking documentation for the below components.

**Neural Network Primitives**   To facilitate the implementation of neural networks, Flashlight ships with common neural building-blocks encompassing activation functions, normalization layers, regularizers, losses, and more. These derive from the MODULE abstraction, as discussed above, which provides a method of chaining and nesting operations.Section A.4.1 contains more details and sample implementations.

**Automatic Differentiation**   Automatic differentiation (autograd) is implemented via a simple VARIABLE[3] abstraction. A Variable takes a TENSOR argument when created, and its underlying Tensor (or gradient Tensor) can be accessed at any time. Variables feature operators which call underlying Tensor operations and record those operations to a dynamic tape in a design similar to Paszke et al. (2017) while being lightweight enough to allow implementations of other autograd paradigms. In keeping with Flashlight's modularity, TENSOR and VARIABLE are separated in order to avoid performance and implementation overhead in non-gradient-based ML algorithms.

```
1 Variable cos(const Variable& input) {
2   auto result = fl::cos(input.tensor()); // get a Tensor from a Variable
3   // Called with backward() to compute gradients for this op's inputs
4   auto gradFunc = [](std::vector<Variable>& inputs,
5                      const Variable& gradOutput) {
6     inputs[0].addGrad( // Add a gradient to the input
7         Variable(gradOutput * negate(sin(inputs[0].tensor())), false));
8   };
9   // Construct a Variable from a Tensor and a gradient-computing function
```

---

[3]redactedforanonymity

```
10    return Variable(result, {input}, gradFunc);
11 }
```

Listing 4: Defining a cosine autograd operator in Flashlight using TENSOR operations and VARIABLE.

**Optimizers**   Flashlight provides implementations of most common first-order stochastic optimization algorithms, as included in other frameworks. These are defined in terms of Variable and Tensor operations, allowing for open-ended experimentation (e.g. with distributed computation, in-place operations, etc).

**Data Loaders**   Flashlight provides a simple DATASET class which abstracts away the notion of a *sample* in ML algorithms. A sample is viewed here as a TENSOR or vector of TENSORS. Datasets are trivially composable to create pipelines to transform, resample, or parallelize (via native C++ threads) the construction of such samples. While Flashlight core dataset abstractions are agnostic from the end-user task, data-specific datasets are provided in higher-level *packages*, to efficiently load from disk structured data (e.g. images, audio or text).

### 4.3   PACKAGES AND APPLICATIONS

Flashlight contains additional domain-specific abstractions leveraging both core components as well as stand-alone implementations. These abstractions allow end-users or ML researchers to quickly get started on various ML applications. The *package* module provides building blocks for common ML tasks, domain-specific algorithms, and helpers. The *application* module leverages these building blocks to provide complete, ready to use solutions (e.g. models, training loops, evaluation pipelines). When not original to Flashlight, implementations reproduce the task performance those they reference. We leverage several of these applications to evaluate Flashlight's performance in Section 5.

**Speech.**   Flashlight provides an implementation of classical featurization (spectogram, log-mel filterbanks, etc.) that can run on-the-fly with minimal overhead. It also provides a collection of data-augmentation techniques (including additive noise and reverberation), as well as implementations of speech-specific sequential criteria and model architectures. Flashlight contains a fast beam-search decoder (which can interface any language model) and beam rescorers (Collobert et al., 2016; Pratap et al., 2019). Research performed with the speech *application* have reached and are competitive with state-of-the-art results (Synnaeve et al., 2019; Likhomanenko et al., 2020).

**Vision.**   Flashlight offers built-in data loaders for standard computer vision benchmarks (such as ImageNet (Deng et al., 2009) and COCO (Lin et al., 2014)) along with large set of efficient data-augmentations and transformations. It includes mainstream image classification models: convolutional (e.g. ResNet He et al. (2016)) and transformer-based architectures (e.g. ViT Dosovitskiy et al. (2020)), as well as a modern, transformer-based object detection model (DETR Carion et al. (2020)) and helpers (e.g. Hungarian matching and object detection evaluation).

**Text.**   Flashlight ships with support for text dataset manipulation and tokenization along with language modeling training pipelines for a variety of neural language models, including transformer (Vaswani et al., 2017) and CNN-based (Dauphin et al., 2017). Both autoregressive and masked, e.g. BERT, language modeling tasks are supported. These language models can be combined with other domain-specific packages such as speech.

## 5   EVALUATION

In the sections that follow, we evaluate Flashlight in context with two widely-used deep learning frameworks — PyTorch and Tensorflow — with metrics relevant to framework research velocity. We also evaluate framework performance as a proxy for overhead and the quality of default implementations. We outline the steps needed to reproduce all our results included in this section in the Appendix.

Table 1: Complexity of various frameworks based on high-level metrics.

| Metric | PyTorch | TensorFlow | (Ours) Flashlight |
|---|---|---|---|
| Binary Size (MB) | 527 | 768 | 9 |
| Lines of Code | 1,798,292 | 1,306,159 | 27,173 |
| Number of Operators | 2,166 | 1,423 | 110 |
| **Approx num. ops. that perform:** | | | |
| ADD | 55 | 20 | 1 |
| CONV | 85 | 30 | 2 |
| SUM | 25 | 10 | 1 |

Table 2: Compile times in CPU minutes across frameworks.

| Platform | From Scratch | Incremental |
|---|---|---|
| PyTorch | 754 | 132 |
| Tensorflow | 2061 | 371 |
| (Ours) Flashlight | 27 | 0.6 |

## 5.1 CODE COMPLEXITY

Flashlight is built to minimize complexity and operating surface. As frameworks grow and are combined with other frameworks or take on new platform-specific requirements, internal modifiability decreases. Table 1 compares frameworks across binary size, lines of code[4], and number of operators and operator implementations. Flashlight's small surface facilitates easily exploring new designs and prototyping on new hardware — having few sources of truth simplifies the process of replacing core components and ensures end-to-end tests don't opaquely fall back to existing implementations.

### 5.1.1 COMPILATION TIME

When modifying or adding significant new research code to framework internals, recompilation can be costly. Large frameworks depend on code generation for broad platform support[5], increasing compilation time. Further, expensive incremental rebuilds can slow iteration speed.

Flashlight is sufficiently-lightweight and modular so as to enable from-source build times that are orders of magnitude faster than other frameworks, as shown in Table 2. Times were measured for both from-scratch and incremental builds with Intel Xeon Gold 6138 CPUs with 80 cores and 750 GB of memory. To estimate incremental build performance, we randomly sample 100 source files without replacement, make trivial modifications that force recompilation, and time the resulting rebuild.

### 5.1.2 PERFORMANCE

When improving framework components or modifying internals, framework overhead makes it difficult to disambiguate performance changes due to in-flight modifications from existing bottlenecks or overhead due to other framework components as discussed in Section 3. Table 3 compares the performance of Flashlight 0.3.1, PyTorch 1.8, TensorFlow 2.4 on 6 common large-scale deep neural networks. For each configuration, we benchmark 100 iterations of data loading[6], preprocessing, and forward/backward passes, with data-parallel gradient synchronization in distributed settings. Benchmarks are performed on Intel E5-2698 CPUs with 512GB of RAM, and NVIDIA V100-32GB GPUs in a DGX-1 Station. Inter-GPU interconnects in the 8 GPUs (1 node) setting and the multi-node (64 GPUs) setting are Nvidia NVLink and InfiniBand-based, respectively.

---

[4]In an attempt to not include auxiliary framework components, we count only C, C++, Python, YAML, CUDA, and CMake files from a relevant subset of core components for each project.

[5]Examples | PyTorch: `https://git.io/Jzel9`, TensorFlow: `https://git.io/JzeRw`

[6]For consistency, BERT-like models use random data in-memory; ViT models exclude data augmentation.

Table 3: Performance on common state-of-the-art models across frameworks. Values are the number of seconds needed to perform 100 iterations of the forward and backwards passes, with data loading (unless indicated). Number of parameters in millions (**#param**) and batch size (**bsz**) are specified. Framework labels: **PT** = PyTorch, **TF** = TensorFlow, and **FL** = Flashlight.

| Model | | | 1 GPU | | | 8 GPUs | | |
|---|---|---|---|---|---|---|---|---|
| | Num. Params (M) | Batch Size | PT | TF | FL | PT | TF | FL |
| AlexNet | 61 | 32 | 2.0 | 4.0 | 1.4 | 6.0 | 6.5 | 2.1 |
| VGG16 | 138 | 32 | 14.8 | 12.6 | 13.2 | 16.3 | 17.9 | 14.9 |
| ResNet-50 | 25 | 32 | 11.1 | 12.4 | 10.3 | 12.3 | 15.9 | 11.9 |
| BERT-like | 406 | 128 | 19.6 | 19.8 | 17.5 | 22.7 | 23.6 | 19.2 |
| ASR Tr. | 263 | 10 | 58.5 | 63.7 | 53.6 | 63.7 | 69.7 | 57.5 |
| ViT | 87 | 128 | 137.8 | 140.3 | 129.3 | 143.1 | 169.6 | 141.0 |

Flashlight is competitive and can exceed the performance of other frameworks, especially on architectures which are of lower arithmetic intensity and spend less compute time in vendor-optimized libraries, such as AlexNet. Given strong performance with simple reference implementations that have undergone far less optimization than have large frameworks, we see exciting potential for improvement with future research done in Flashlight.

## 6 CONCLUSION

We presented Flashlight, a modular machine learning library supporting modern, state-of-the-art baselines that features orders of magnitude less code and binary size as compared to frameworks such as PyTorch and TensorFlow. These large frameworks come fully-featured; Flashlight aims to complement them in providing a tool to do machine learning framework research. To this end, Flashlight features a lightweight, modular design, as well as full implementations of mainstream models across a variety of domains, making it easy for researchers to implement new internal tensor, memory management, or distributed computation backends. Flashlight includes a variety of reference APIs that compete with and often outperforms that of popular machine learning frameworks, thus demonstrating the viability of our approach.

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

## A  APPENDIX

### A.1  REPRODUCIBILITY

All tools and code used in our evaluation are open source and available on Github — this includes scripts to reproduce benchmarks and other quantitative codebase metrics. Code can be found at *URL redacted to preserve anonymity*.

Flashlight is open source on Github[7] as stated in Section 5, Flashlight v0.3.1 is used to reproduce results using ArrayFire 3.8 as the underlying tensor backend. No other specialized configuration is used for either Flashlight or PyTorch or TensorFlow.

#### A.1.1  FROM-SOURCE COMPILATION

Flashlight is built in CMake release mode, which is also the default for both PyTorch and TensorFlow builds. Build settings are kept default for all frameworks; Flashlight uses Ninja[8] with CMake in accordance with PyTorch's build, while Tensorflow employs Bazel.

Exact build step reproduction can be found in the aforementioned repository.

#### A.1.2  INCREMENTAL COMPILATION

Incremental compilation benchmarks are performed using similar build setups per Section A.1.1. To test incremental rebuilds, source files are randomly selected without replacement from a distribution constructing by weighting each file by the number of lines in the file, and using that number to determine the probability of selecting it for modification.

Scripts to perform and time this incremental compilation can be found in the aforementioned public repository.

### A.2  CODE COMPLEXITY

#### A.2.1  OPERATOR COUNTING

To count the number of operators for each framework, we utilize operator schemas for PyTorch and Tensorflow (which generate code from those schemas, accordingly) written in YAML and Protobuf, respectively. For Flashlight, we count the number of functions in the Flashlight TENSOR interface and autograd interfaces, as these form the full implementation requirements for a full tensor backend that functions on all platforms. The scripts released on Github detail the files and filtering techniques used to reproduce the number of results.

To count the number of operators for each implementation, we use the above operator lists, then count the number of operators that perform the specified function, even if those operators perform other functions. For example: an operator called ADDMM, which performs an addition operation followed

---

[7]`redactedforannonymity`
[8]https://ninja-build.org/

by a matrix-matrix multiplication, performs an addition operation, and would thus be counted when tallying the number of ADD operators.

## A.3 PERFORMANCE

The aforementioned public repository provides scripts required to fully-reproduce all performance measures.

## A.4 DESIGN DETAILS AND CODE SAMPLES

In the following sections, we show brief code samples expounding on those in Section 4.

### A.4.1 MODULES

Flashlight's MODULE abstraction is similar to that of frameworks such as Torch and PyTorch. It can recursively store other modules and interoperate with more sophisticated abstractions including CONTAINER, which wraps multiple modules, SEQUENTIAL, which stores sequences of modules and forwards data through them sequentially, and user-defined abstractions. Listing 7 in Section A.4.2 shows an example of Sequential usage.

Listing 5 shows a small Dropout module implementation that calls into the dropout autograd primitive, stores and serializes a small amount of state, and defines a simple forward function on a Variable.

```
1  class Dropout : public Module {
2   private:
3     double ratio_;
4     FL_SAVE_LOAD_WITH_BASE(Module, ratio_) // serialization
5
6   public:
7     Dropout(double drop_ratio = 0.5);
8     Variable forward(const Variable& input) override {
9       if (train_) {
10         return dropout(input, ratio_); // autograd primitive
11       } else {
12         return input;
13       }
14     }
15     // ...
16  };
```

Listing 5: A Dropout layer implemented as a Flashlight module.

The *FL_SAVE_LOAD_WITH_BASE* macro defines serialization of the Dropout class as a module, including any fields to be serialized (in this case, only the dropout ratio).

### A.4.2 AN END-TO-END EXAMPLE: MNIST

Below, we detail a simple end-to-end training setup.

First, data is loaded using the BATCHDATASET abstraction in Listing 6:

```
1  const int kTrainSize = 60000;
2  const int kValSize = 5000;
3
4  auto& [train_x, train_y] = load_dataset(data_dir);
5
6  // Hold out a dev set
7  auto val_x = train_x(span, span, range(0, kValSize));
8  train_x = train_x(span, span, range(kValSize, kTrainSize));
9  auto val_y = train_y(range(0, kValSize));
10 train_y = train_y(range(kValSize, kTrainSize));
11
12 // Make the training batch dataset
13 BatchDataset trainset(
```

```
14      std::make_shared<TensorDataset>(std::vector<Tensor>{train_x, train_y
        }),
15      batch_size);
16
17 // Make the validation batch dataset
18 BatchDataset valset(
19      std::make_shared<TensorDataset>(std::vector<Tensor>{val_x, val_y}),
20      batch_size);
```

Listing 6: Loading MNIST data into a train and evaluation set.

A full description of the LOAD_DATASET function can be found in the MNIST training example on Github[9].

We can construct the model using a simple SEQUENTIAL in Listing 7:

```
1 const int kImageDim = 28;
2 auto pad = PaddingMode::SAME;
3
4 Sequential model;
5 model.add(View({kImageDim, kImageDim, 1, -1})); // WHCN (col major)
6 model.add(Conv2D(
7     1 /* input channels */,
8     32 /* output channels */,
9     5 /* kernel width */,
10    5 /* kernel height */,
11    1 /* stride x */,
12    1 /* stride y */,
13    pad /* padding mode */,
14    pad /* padding mode */));
15 model.add(ReLU());
16 model.add(Pool2D(
17    2 /* kernel width */,
18    2 /* kernel height */,
19    2 /* stride x */,
20    2 /* stride y */));
21 model.add(Conv2D(32, 64, 5, 5, 1, 1, pad, pad));
22 model.add(ReLU());
23 model.add(Pool2D(2, 2, 2, 2));
24 model.add(View({7 * 7 * 64, -1}));
25 model.add(Linear(7 * 7 * 64, 1024));
26 model.add(ReLU());
27 model.add(Dropout(0.5));
28 model.add(Linear(1024, 10));
29 model.add(LogSoftmax());
```

Listing 7: Constructing a CNN for MNIST training.

In Listing 8, we create a simple custom training loop. This uses optimizer, loss function, and meter abstractions as provided by default by Flashlight. We perform the forward and backward pass, step the optimizer to update parameters, and zero out gradients before moving to the next batch. We evaluate the model using the function defined in Listing 9, pulls out the max prediction and comparing it against the ground truth, updating the loss meter as we go, then returning the final loss values.

```
1 // Make the optimizer
2 SGDOptimizer opt(model.params(), learning_rate);
3
4 // The main training loop
5 for (int e = 0; e < epochs; e++) {
6   AverageValueMeter train_loss_meter;
7
8   // Get an iterator over the data
9   for (auto& example : dataset) {
10     auto inputs = noGrad(example[INPUT_IDX]);
```

---

[9]redactedforanonymity

```
11      auto output = model(inputs);
12
13
14      auto target = noGrad(example[TARGET_IDX]);
15
16      // Compute and record the loss.
17      auto loss = categoricalCrossEntropy(output, target);
18      train_loss_meter.add(loss.tensor().scalar<float>());
19
20      // Backprop, update the weights and then zero the gradients.
21      loss.backward();
22      opt.step();
23      opt.zeroGrad();
24    }
25
26    double train_loss = train_loss_meter.value()(0).scalar<double>();
27
28    // Evaluate on the dev set.
29    double val_loss, val_error;
30    std::tie(val_loss, val_error) = eval_loop(model, valset);
31
32    std::cout << "Epoch " << e << std::setprecision(3)
33             << ": Avg Train Loss: " << train_loss
34             << " Validation Loss: " << val_loss
35             << " Validation Error (%): " << val_error << std::endl;
36 }
```

Listing 8: A simple training loop.

```
1 std::pair<double, double> eval_loop(Sequential& model, BatchDataset&
      dataset) {
2   AverageValueMeter loss_meter;
3   FrameErrorMeter error_meter;
4
5   // Place the model in eval mode.
6   model.eval();
7   for (auto& example : dataset) {
8     auto inputs = noGrad(example[INPUT_IDX]);
9     auto output = model(inputs);
10
11    // Get the predictions in max_ids
12    Tensor max_vals, max_ids;
13    max(max_vals, max_ids, output.tensor(), 0);
14
15    auto target = noGrad(example[TARGET_IDX]);
16
17    // Compute and record the prediction error.
18    error_meter.add(reorder(max_ids, 1, 0), target.tensor());
19
20    // Compute and record the loss.
21    auto loss = categoricalCrossEntropy(output, target);
22    loss_meter.add(loss.tensor().scalar<float>());
23  }
24  // Place the model back into train mode.
25  model.train();
26
27  double error = error_meter.value().scalar<double>();
28  double loss = loss_meter.value().scalar<double>();
29  return std::make_pair(loss, error);
30 }
```

Listing 9: Evaluating a training model on MNIST.

Finally, in Listing 10, we evaluate the trained model on the test set by creating a test dataset and using the previously-defined evaluation function.

```
1  std::pair<double, double> eval_loop(Sequential& model, BatchDataset&
       dataset) {
2    AverageValueMeter loss_meter;
3    FrameErrorMeter error_meter;
4
5    // Place the model in eval mode.
6    model.eval();
7    for (auto& example : dataset) {
8      auto inputs = noGrad(example[INPUT_IDX]);
9      auto output = model(inputs);
10
11     // Get the predictions in max_ids
12     Tensor max_vals, max_ids;
13     max(max_vals, max_ids, output.tensor(), 0);
14
15     auto target = noGrad(example[TARGET_IDX]);
16
17     // Compute and record the prediction error.
18     error_meter.add(reorder(max_ids, 1, 0), target.tensor());
19
20     // Compute and record the loss.
21     auto loss = categoricalCrossEntropy(output, target);
22     loss_meter.add(loss.tensor().scalar<float>());
23   }
24   // Place the model back into train mode.
25   model.train();
26
27   double error = error_meter.value().scalar<double>();
28   double loss = loss_meter.value().scalar<double>();
29   return std::make_pair(loss, error);
30 }
```

Listing 10: Evaluating a training model on MNIST.

