# OpenReview forum: "Flashlight: Enabling Innovation in Tools for Machine Learning"
_ICLR.cc/2022/Conference — ICLR 2022 Submitted_

### Official Review · Reviewer_DYUa · 2021-11-02

**Correctness:** 3
**Technical Novelty And Significance:** 2
**Empirical Novelty And Significance:** 1
**Recommendation:** 5
**Confidence:** 2

**Main Review:**


**Strength of the work.**
- The paper provides performance comparison for end-to-end training performance on 6 representative
workloads, and the proposed system outperforms PyTorch and TensorFlow in most of the cases;
- The proposed system allows customized memory manager, which is a missing piece in many of the existing
deep learning frameworks;
- The system is ready to use in production, which makes it outstanding given the robustness requirement
in the production environment.

**Weakness.**
- Ablation study is missing from the paper. For example, training ResNet-50 on a single GPU
has become a well-studied and highly-optimized task. However, the system is still able to outperform
existing large systems like TensorFlow. Therefore, it would be desirable to showcase why the system
is able to do better. Is it because Flashlight does better latency hiding in data loading, or the
low-overhead nature of C++ APIs, or any other reasons?
- More direct and fair comparison might be required for the reviewer to better understand
in Table 1 and 2. According to Table 1, FlashLight has far fewer operators than PyTorch (110 vs 2166),
so it seems nature that it compiles faster than PyTorch (27min vs 754min), and generates smaller binaries (9MB vs 527MB).
- Claims need more justification with data and comparison with other frameworks. For example, in Section 3,
it's claimed that the proposed system is research-first. It would be desirable to quantifiably understand the reason
behind the claim. Compared with TensorFlow where a lot of research have been done on for distributed training,
why the proposed system is more suitable?

**Correctness.** To the best of the reviewer's knowledge, there is no correctness issue.

**Clarity.** The paper is not hard to read, but might need more data to justify many of it claims.



**Summary Of The Paper:**

The paper presents Flashlight, a C++-based deep learning framework, based on the ArrayFire
tensor library, and implemented all the necessary functionalities in the core infrastructure,
including memory management, distributed training, data loader, etc. The major contribution includes:
- An end-to-end framework with solid engineering efforts, which comes with highly optimized models
including speech, vision and text;
- The system is modularized and customizable with simple C++ APIs, and each component is equipped
with highly optimized default implementation;
- The system is benchmarked with a variety of the state-of-the-art models, and supports end-to-end
packaging for these applications.


**Summary Of The Review:**

In summary, the paper presents a solid deep learning framework written in C++, and has been put in production for a wide range of applications. However, in terms of paper writing, it would be more desirable for the readers and reviewers to understand the novelty and quantifiably justify the claims in the paper.

---

> ### Author Response · Authors · 2021-11-14
> **Response to Reviewer DYUa**
>
> Thank you for your comments. Our detailed responses are below.
>
> > [Q1] FlashLight has far fewer operators than PyTorch (110 vs 2166), so it seems nature that it compiles faster than PyTorch (27min vs 754min), and generates smaller binaries (9MB vs 527MB)
>
> We agree — this is an important part of our premise. Having fewer operators makes frameworks smaller, faster to build, and easy to do tensor and operator-level research with. In Flashlight's case, operators are not composed and are higher-level, enabling the same level of expressiveness and similar or better performance compared to PyTorch and Tensorflow but without added complexity that makes systems research challenging in other frameworks. We argue in Section 5 that the number of operators is inversely proportional to the ease of making wide-ranging modifications to frameworks' tensor internals.
>
> > [Q2] Claims need more justification with data and comparison with other frameworks [...] Compared with TensorFlow where a lot of research have been done on for distributed training, why the proposed system is more suitable?
>
> Section 5 contains many of these justifications on what research-first means in a context of a framework. We welcome further feedback on examples of these claims beyond the one provided, and will update the manuscript to more details on Flashlight's distributed training API and its competencies.
>
> Flashlight's distributed computation API is unopinionated across computation model (sync, async, etc). While research headway has been made with frameworks such as TensorFlow per modifying computation graphs, these modifications are quite limited. Flashlight's distributed computation API is most powerful when used in conjunction with a custom tensor implementation, to relax assumptions around Tensors' device locations and enabling arbitrary or self-optimizing computation models that are opaque to the user. One can imagine a *generalized* paradigm for distributed computation enabled by such flexibility. Further, Flashlight's distributed API can be extended to add new operations with minimal engineering burden, in contrast to other frameworks which feature statically-defined or symbol-based APIs.
>
> > [Q3] Ablation study is missing from the paper
>
> We thank the reviewer for this feedback; while in-depth performance analysis is not the primary goal of our work, we have included more information about understanding system performance in the response to all reviewers above.

---

### Official Review · Reviewer_1j8s · 2021-11-02

**Correctness:** 4
**Technical Novelty And Significance:** 2
**Empirical Novelty And Significance:** 2
**Recommendation:** 3
**Confidence:** 4

**Main Review:**

Flashlight’s narrow use-case, as a testbed for systems research in machine learning, is important and poorly supported by its major competitors. This comes at the cost of other use-cases like model design research (limited feature set, need to compile models in C++) or usage in production environments (avoidance of low-level optimizations, absence of model servers for deployment).

While Flashlight with its native support for efficient memory management, distributed training and end to end benchmarking, shows promise as a research tool. However, it has not demonstrated its capabilities in practice. Whereas a long list of well-established (reference) models in speech, vision, and text applications are provided, the only discussion of novel work enabled by Flashlight is a brief mention of wav2letter in Section 3. The performance study on well-established models gives confidence in the basic implementation of Flashlight, which is of limited use by itself. This would be a strong paper if it contained a case study showing how its design contributed to systems research.


**Summary Of The Paper:**

The paper describes the design philosophy and structure of the Flashlight deep learning framework. Flashlight is modular, small, and narrowly oriented toward systems researchers. Rather than (or in addition to) high level productivity, Flashlight focuses on internal and external simplicity of ML tools. The authors evaluate Flashlight by training several standard reference models and it achieves slightly better performance than PyTorch or TensorFlow


**Summary Of The Review:**

Flashlight is well-designed and shows promise as a research tool for systems research in machine learning. However, it has not demonstrated its usefulness in practice.

---

> ### Author Response · Authors · 2021-11-14
> **Response to Reviewer 1j8s**
>
> Thank you for your comments. Our detailed responses are below.
>
> > [Q1] This would be a strong paper if it contained a case study showing how its design contributed to systems research.
>
> We appreciate this input — please see the "Case Studies" section under our response to all reviewers. We can and will expound on one or more of these case studies in the manuscript detailing the research and downstream framework advances made possible by Flashlight.
>
> As discussed above, we are aware of a wealth of ongoing work that uses Flashlight from organizations including Apple, NVIDIA, Facebook, Google, Stanford, MIT, and SambaNova. We expect those authors to publish and open source their contributions. We hope this addresses the aforementioned concerns.

---

### Official Review · Reviewer_gStW · 2021-11-03

**Correctness:** 3
**Technical Novelty And Significance:** 3
**Empirical Novelty And Significance:** 3
**Recommendation:** 6
**Confidence:** 4

**Main Review:**

The authors have built what appears to be a well thought out, and cleanly implemented neural network framework with researcher modification and extensibility in mind. The paper is well written, with clear background sections on the current state of framework design, and the need for a more research-oriented extensible framework. Indeed a simplistic, easily modifiable/extensible framework that can be used to easily implement new research methods that might require custom forms of memory management, runtime, or tensor functions is immediately appealing.

While I commend the authors for what is no small undertaking, I have a few concerns:

1. The first and primary one is whether this work sufficiently distinguishes itself in a crowded field. Between the PyTorch C++ library, Tensorflow Lite, Tensorflow Lite Micro, etc., there are many options for implementing custom built solutions, some of which offer a similar level of control over memory management, and operator behavior as described in Flashlight (albeit somewhat less cleanly perhaps). Via the PyTorch C++ library, custom operators can be implemented, memory allocation behavior can be customized as desired, and hooking into the PyTorch ecosystem carries significant advantages. So the question I'm left asking myself is whether Flashlight provides any significant advantages over the PyTorch C++ library for the majority of use cases. Unless a researcher is implementing functionality that really does involve altering very large portions of framework internals, I'm not convinced that optimizations are worth giving up for the sake of readability.

That being said, the framework seems nicely designed, with clean interfaces and its use of ArrayFire for (nice) cross-platform tensor calculation, and a small binary size and compile time to boot which is great for development purposes (although again, extending PyTorch with a custom op does not require compiling the library from scratch). I could see using Flashlight in order to roll a very custom solution.

I am willing to believe that the code cleanliness, modularity and practicality of Flashlight could be beneficial to some researchers, but beyond that, I'm unsure as to what Flashlight brings to the table other than that it's more barebones and a good jumping off point for research than other frameworks. The methods it employs are (unless I'm mistaken) not exactly unique to the work, and I'm unsure if a conference paper is necessarily the right venue for this work.

2. I assume the 9MB binary size stated in the paper does not include ArrayFire... Does the PyTorch 528mb include Aten? and similar for Tensorflow?

3. I would like to hear more about why Flashlight performs better than PyTorch or Tensorflow. Is this an Aten v.s. ArrayFire speedup?

**Summary Of The Paper:**

This paper proposes a neural network training and inference framework aimed at framework researchers with a focus on modularity, simplicity of design, and extensability by researchers.

**Summary Of The Review:**

Overall the paper was interesting, and I commend the authors for their work. It seems the Flashlight framework could be beneficial for researchers looking for a boilerplate template framework on which to build their very highly customized solutions. However, for anything other than an solution requiring rewriting large portions of a neural network framework, the researchers using this library will give up optimizations and other benefits of using a more mature library such as PyTorch C++ (for which custom ops and some other features mentioned in this work are possible without rewriting internals).

---

> ### Author Response · Authors · 2021-11-14
> **Response to Reviewer gStW**
>
> Thank you for your comments. Our detailed responses are below.
>
> > [Q1] for anything other than an solution requiring rewriting large portions of a neural network framework, the researchers using this library will give up optimizations and other benefits of using a more mature library
>
> We agree with this characterization — Flashlight targets research at the computational foundations of machine and deep learning. While it has enabled a variety of pure and applied research in multiple domains, as discussed in Section 3, Flashlight foregoes some optimizations for specific research use cases to keep the framework lightweight.
>
> > [Q2] memory allocation behavior can be customized as desired [...] custom operators can be implemented
>
> There exists no universal, cross-platform interface in PyTorch (polymorphic, symbol-based, or otherwise) with which to customize memory managers beyond changing the source per the default memory manager. While this has been discussed extensively [1], framework complexity and numerous assumptions that the memory manager makes seem to be the bottleneck to adding such an interface; these are expressly-mitigated by Flashlight's design. Ultimately, we argue that having an interface that is built explicitly for research and extension is very different from modifying or hacking on internal components not meant for experimentation and is far faster and less error-prone.
>
> While it is true that custom operators can be implemented across most frameworks, Flashlight enables research in several important directions that are not possible in other frameworks, as outlined in Section 4.1.1. These may include:
> 1. New computation models. Flashlight's tensor API is agnostic to computation model (such as lazy code generation, eager, static, or something novel as in Figure 2), whereas implementing tensor backends in PyTorch or TensorFlow require eager or static implementations, respectively.
> 2. Other frameworks make changing existing operator implementations extremely difficult (requiring source modification), and only provide APIs for adding custom operators, which aren't leveraged in existing downstream models without significant code changes, i.e. swaping to use those custom operators. As an example: this is the difference between adding a special convolution operator then changing numerous downstream projects to use it versus changing the core, default convolution operator that's already-used downstream.
> 3. Changing behavior of a single, simple operation (e.g. sum, convolution) to explore operator efficiency may require changing dozens of places in which custom code performing that operation is used and with slightly different behavior, as described in the bottom of Table 1.
> 4. Broad, holistic changes to how tensor computation is done (e.g. new end-to-end compiler techniques) across a framework requires modifying all usable operators, which in large frameworks, requires changing thousands of callsites (see Table 1).
>
> We thank the reviewer for this feedback and will clarify these distinguishing points in the manuscript.
>
> > [Q3] I assume the 9MB binary size stated in the paper does not include ArrayFire... Does the PyTorch 528mb include Aten? and similar for Tensorflow?
>
> We appreciate this feedback and this is well taken — the PyTorch and Tensoflow figures include tensor libraries. We will update the figure to exclude tensor stacks from each framework excluding ATen for PyTorch and Tensorflow ops, XLA, and JIT components. For Tensorflow, we compute binary size by individually-compiling each of the subtracted components above and subtracting their size from overall binary size. To our knowledge, it is not possible to compile PyTorch without ATen — we will clearly indicate this per the comparison. Framework size and lines of code remains unchanged for Flashlight, but updates values for other frameworks become:
>
> | Metric              | PyTorch          | Tensorflow | Flashlight (ours) |
> |---------------------|------------------|------------|-------------------|
> | Binary Size (MB) (no tensor lib)    | (*) | 423        | 9                 |
> | Lines of Code (no tensor lib)       | 924K             | 604K       | 27K               |
> | Binary Size (MB) (with tensor lib)    | 527 | 768        | 139^                 |
> | Lines of Code (with tensor lib)       | 1798K             | 1306K       | 97K^               |
> | Number of Operators | 2,166            | 1,423      | 110               |
>
> *PyTorch cannot be compiled without ATen.
>
> ^Using Flashlight with ArrayFire. Flashlight can be compiled with smaller tensor libs as well.
>
> We will update the manuscript with these results and provide the above and more details.
>
> > [Q4] I would like to hear more about why Flashlight performs better than PyTorch or Tensorflow.
>
> We thank the reviewer for this feedback; we have included more information about understanding system performance in the response to all reviewers.
>
> [1] https://github.com/pytorch/pytorch/issues/43144

---

### Official Review · Reviewer_KzrA · 2021-11-05

**Correctness:** 2
**Technical Novelty And Significance:** 3
**Empirical Novelty And Significance:** 2
**Recommendation:** 5
**Confidence:** 3

**Main Review:**

I have to admit that this paper gives me mixed feelings. Machine learning compiler/framework is my major of study, and I understand how we often have the desire to redesign the framework. It is not only (in most of the cases) harder than anticipated, but also (in most cases) just create "another" framework that is not necessary better. It is also very hard to argue that a framework is better, since there isn't a good way to measure the "quality" of a framework.

Strengths:
1. Flashlight captured the "core" of the Machine Learning Frameworks: Tensor and Operation, Memory Management, and Distributed Computation. The core APIs are interesting and somewhat innovative.


Weakness:
1. I have to admit that it is tremendously hard to argue (with evaluation) that one machine learning framework is better than another. In this paper, we see too many vague arguments such as "agile", "modular", and "nimble". It is unfortunately that such vague arguments are bad for a scientific paper. On the same cord, the evaluations in the paper doesn't really mean too much. For instance, Table 1 argues that Flashlight is small. Table 2 argues that Flashlight is quick to compile. Table 3 show model performance, but most models are similar (AlexNet is small and small models are easier to be faster in a simple framework). None of the evaluations support the argument of "agile", "modular", and "nimble".

2. The interesting parts (or, we could say the "pain points" of machine learning frameworks in general) are memory management and distributed computation. The paper highlighted the importance of them, but the contribution of Flashlight in these aspects seems minimal (just described the minimal APIs and connected to downstream implementations). I agree that other researchers might be able to take advantage of these APIs, but the contribution seems small.

How could the paper be improved?

Honest, I am not sure. To be fair, Pytorch paper was published years after it was popular. TensorFlow was published when it was already well-known. Recent, MLIR made a big noise in machine learning framework community, but still, the popularity of the tool is way more important that the publication of it.

Taking MLIR as a reference, I think the reason it is loved so much is that it facilities (greatly) the development of IRs and IR transformations. I think it will be helpful to show case how Flashlight facilities other researchers by the "agile and minimal" design. That would be a more convincing argument of the value of Flashlight.



**Summary Of The Paper:**

The paper proposed a minimal design API (or mostly API?) of machine learning framework called Flashlight. The key argument of the paper is that Flashlight is modular and agile. The Flashlight captured the key aspects of machine learning frameworks: Tensor and Operation, Memory Management, and Distributed. There are some evaluations to argue the benefit of Flashlight.

**Summary Of The Review:**

I empathy the difficult of arguing the "betterness" of a machine learning framework. However, I think the paper in general has not made a convincing argument or provide enough scientific contribution. So I recommend rejecting this paper. However, I am very interested to check out the open source implementation of Flashlight, and see how amazing it is.

---

> ### Author Response · Authors · 2021-11-14
> **Response to Reviewer KzrA**
>
> Thank you for your comments. Our detailed responses are below.
>
> > [Q1] [Redesigning a framework] just create "another" framework that is not necessary better
>
> We disagree with this characterization of our work. Our assertion is not that Flashlight is universally better — it is different and serves a different purpose. We agree with the reviewer in that framework redesign is difficult and believe this further supports the importance of a tool that can be used to experiment with framework design and computation paradigms — this is at the core of Flashlight's purpose, as discussed in the manuscript. As argued: it was software and framework innovation that partially enabled the growth and scale of deep learning-based approaches.
>
> > [Q2] there isn't a good way to measure the "quality" of a framework
>
> We disagree, and set out to give important metrics for systems and framework research velocity in Section 5 of the manuscript including (1) codebase size, (2) number of sources of truth, (3) build time, and (4) flexibility and openness of internal APIs. Measuring framework quality is complex and depends on user needs, but to say that frameworks cannot be better than one another for paticular use-cases makes explaining the growth of certain frameworks (i.e. PyTorch and Tensorflow) impossible. We are open to specific suggestions as to other metrics that might be relevant to evaluating systems and framework research productivity or to feedback on our justifications of included metrics' importance.
>
> > [Q3] too many vague arguments such as "agile", "modular", and "nimble" [...] None of the evaluations support the argument of "agile", "modular", and "nimble".
>
> The word "agile" does not appear in the manuscript. The word "nimble" appears once when describing research, rather than Flashlight as a framework. Modularity is a well-established design principle in research and engineering [1] and codebase size is a well-established metric for software complexity [2], as is functional duplication of code (i.e. operators performing similar functions) [3]. We refer the reviewer to Section 5, which argues that operator quantity and complexity is important for research on computational foundations. We can and will update the manuscript to cite these and other sources supporting the objectivity of our measurements in other domains outside of machine learning. As before, we welcome specific feedback on our assessments and use of these metrics.
>
> > [Q4] The interesting parts (or, we could say the "pain points" of machine learning frameworks in general) are memory management and distributed computation
>
> While memory management and distributed computation are indeed important, we argue that advances in tensor computation (e.g. cuDNN, XLA, Halide, TVM, XLA and others as cited) have also vastly changed the research landscape enabling significant performance improvements, research directions, and portability to new accelerators and hardware. We believe it essential that research into these areas be accelerated. Section 1 describes the significant need and empirical importance of such advances.
>
> > [Q5] the popularity of the tool is way more important that the publication of it
>
> While we respectfully disagree with this premise, Flashlight is fundamentally different than PyTorch and Tensorflow and serves a different purpose. As discussed above, it is not a general-use framework that we are arguing is universally better — Flashlight aims to enable high velocity systems and framework-level research in ways other frameworks cannot. We argue that these research directions are critical in Section 1. In this way, Flashlight will not be broadly popular for machine learning practicioners as much as it will be a tool for framework and systems-level research, which has a smaller base of researchers; we believe an apples-to-apples comparison to other frameworks is unfair.
>
> > [Q6] I think it will be helpful to show case how Flashlight facilities other researchers
>
> We appreciate this input — please see the "On External Usage" section under our general response to reviewers. As stated in the top-level reviewer comments, we already see increased interest in Flashlight and popularity across hardware vendors and contributions from researchers at Apple, NVIDIA, Facebook, Google, Stanford, MIT, SambaNova, and others.
>
> [1] https://www.semanticscholar.org/paper/On-the-Modularity-of-Software-Architectures%3A-A-Sant'Anna-Figueiredo/ee0d0e6884dc3ef7342b5366d803c6dd051633a0
>
> [2] https://www.semanticscholar.org/paper/Measuring-software-design-complexity-Card-Agresti/37bf1ec034ab53bf3d38c5708e13527c330c69f2
>
> [3] https://ieeexplore.ieee.org/abstract/document/514697

---

### Author Response · Authors · 2021-11-14
**Top-Level Response to All Reviewers**

Dear reviewers,

Thank you for your constructive feedback. We appreciate the positive feedback including comments such as: *[Flashlight is] a simplistic, easily modifiable/extensible framework that can be used to easily implement new research methods that might require custom forms of memory management, runtime, or tensor functions is immediately appealing* (reviewer gStW); *Flashlight is well-designed and shows promise as a research tool for systems research in machine learning* (reviewer 1j8s); and that *[Flashlight's] use-case, as a testbed for systems research in machine learning, is important and poorly supported by its major competitors* (reviewer 1j8s).

We start by addressing feedback that was common amongst reviewers.

### (1) On External Usage

Flashlight is already in active use by and features contributions from researchers at Apple, NVIDIA, Facebook, Google, Stanford, MIT, SambaNova, and others.

Ongoing research efforts using Flashlight as a starting point include work in code generation, compilers and IRs, memory management, and distributed computing. These are projects that are uniquely-enabled by Flashlight's core competencies; systems and framework researchers are using Flashlight per other frameworks' inabilities to tool their research needs.

### (2) On Case Studies

We appreciate and agree with multiple reviewers' assessments that a case study would strengthen the paper significantly. We will add a real-world case study to the manuscript. Below, we briefly describe two early, recent case studies which may be of interest:

#### (A) Fragmentation Reduction via Research Enabled by Flashlight's Memory Management Interface

While those researching memory management techniques can make ad-hoc modifications to other frameworks' memory managers, build time, internal complexity, and lack of a unified interface makes this challenging. Flashlight was used to research and develop new techniques for fragmentation reduction for GPU memory management. This was made possible by the ease of extending a lightweight memory manager interface in Flashlight along with clear implementation requirements and tests that made rapid prototyping possible. These techniques were tested on a variety of models in Flashlight before researchers shared their findings with the PyTorch team.

This work was eventually downstreamed into PyTorch and resulted in lower fragmentation and more efficient memory usage for large models across accelerators in the PyTorch ecosystem.

#### (B) Differentiable Beam Search Decoding

The ability to change the assumptions behind Flashlight's tensor and automatic differentiation (autograd) components via extensible APIs facilitated research resulting in a fully differentiable beam search decoder, which required operating on unconventional computation graphs not supported by other frameworks' autograd systems.

#### (C) Future Work

In similar veins to the aforementioned ongoing research using Flashlight, we further motivate research enabled by Flashlight's access to tensor stack internals.

Specific measurement of operator performance in machine learning and deep learning is critical to maximize hardware utilization; many frameworks, however, only provide high-level metrics (e.g. timing and memory usage), and a large operator surface makes broad, in-depth analysis difficult. Several current research directions using Flashlight involve (1) building detailed cost models of operator performance using low-level interoperability, then (2) dynamically reorganizing or refining computation on-the-fly for improved performance. This approach is possible in both training and inference settings with Flashlight, and can help guide other frameworks forwards.

### (2) Understanding Performance

Multiple reviewers expressed interest in a more detailed performance study of Flashlight. While not a primary goal of our work, we provide more details below, and will include as many of these details as possible in the paper and appendix, space permitting.

A large part of the performance gap between Flashlight and other frameworks (70-80% for most models) exists due to overhead in dataloading that we attribute to contention inside interpreters with locks around resources shared across threads. Flashlight's shedding the interpreter and using native threading removes these limiting factors. We have also performed additional benchmarks without data loading in which framework performance is relatively similar across the board.

In several cases, even without data loading, Flashlight still outperforms these frameworks for smaller models; we attribute this to low framework overhead and efficient just-in-time compilation. Indeed, as accelerators become more computationally-powerful, framework overhead will become a significant factor in overall performance. We see that on faster accelerators (i.e. NVIDIA A100 versus V100 GPUs), the performance gap with Flashlight widens, even without dataloading.

---

### Author Response · Authors · 2021-11-30
**Follow-Up Response to All Reviewers**

## Additional Details on Case Studies

We are providing additional technical details about the [aforementioned case studies](https://openreview.net/forum?id=C4o-EEUx-6&noteId=MTnRIh3zOr4) per a reviewer's suggestion and interest. These may be relevant to other reviewers per prior feedback.

### More Details: Differentiable Decoding and Custom Autograd Implementations

The researchers building a differentiable beam search decoder encountered several issues necessitating building a custom Autograd implementation in Flashlight:
- A huge autograd graph (with up to millions of operations) that created significant memory pressure
- Small operator overhead per autograd graph node (many addition and log operations)
- These inexpensive operations in the autograd graph had few opportunities for vectorization
- The components of the autograd graph that were used were quite sparse

Based on their writing, researchers modified Flashlight's autograd to support:
- On-the-fly graph pruning to take advantage of sparsity and reduce memory footprint
- Dynamic, pre-fused gradient computation for common sequences of gradient computation operations
- Customizations enabling custom autograd node lifetime so as to avoid shared pointer overhead for large graph mutations

To our knowledge, these capabilities only exist in autograd implementations like Flashlight's that feature public APIs built for customization. Other frameworks' autograd implementations are hidden behind abstractions users are not meant to break open.

### More Details: on Memory Management Fragmentation Reduction

Various caching memory allocator  are used across deep learning frameworks to reduce the cost of native device memory allocations and reuse already-allocated memory. As is the case with other memory management schemes, these caching memory managers are subject to internal and external fragmentation as they bucket allocations based on size rounding. Reducing this fragmentation allows for training larger models and also significantly improves performance, as native memory operations on accelerators typically block computation streams.

Researchers aiming to reduce external fragmentation implemented a custom caching memory manager to study memory behavior and built highly-specialized telemetry that tied individual tensor operations to specific allocations. This telemetry required detailed memory traces and required a customized memory manager implementation. Researchers detailed a myriad of prototypes to reduce fragmentation and described rapid rebuilds and specialized end-to-end measurements as critical to thier experimentation.

A custom memory manager that prevented large cached memory blocks from being split ultimately showed promise. Researchers creating a new parameter: the maximum block size which was eligible for splitting if needed. This reduced fragmentation caused by splitting large cache blocks for significantly-smaller allocations and improved both internal and external fragmentation.

The below figure demonstrates how this modification (using a 256 MB split size limit) significantly reduced the number of malloc/free calls and prevents some OOMs due to reduced fragmentation, significantly boosting performance. Researchers verified these results on NVIDIA GPUs and shared findings with the PyTorch team, which reworked portions of their memory management system to enforce a maximum cache block size limit. These results are shown for a ResNet-50 model with random inputs, and extended to transformer-based models and other architectures.

---

### Decision · Program_Chairs · 2022-01-20

**Decision:**

Reject

**Comment:**

This paper describes Flashlight, a tool for ML researchers with specific design considerations for conducting systems research. The needs for such tool are significant, and recent advances in this topic have been relatively slow, so this research is timely and important.

Reviewers are positive about the importance of the problem and the nice design of Flashlight. It seems the tool has been used by researchers with positive feedback. At the time of the original submission, reviewers expressed some concerns about the novelty and the weak arguments for convincingly showing the advantages over other similar tools.

Authors provided nice replies including specific case studies, but with the short time period to reassess the proposed changes and additions, some reviewers remain hesitant, and thus this paper cannot be accepted at this time. I strongly encourage the authors to incorporate all of the proposed revisions and resubmit to a future venue.